# Assessment of Luteal Function Using Rectal Palpation, B-Mode Ultrasonography, and Progesterone Determination to Improve Recipient Selection in Embryo Transfer Programs

**DOI:** 10.3390/ani13182865

**Published:** 2023-09-09

**Authors:** Uxía Yáñez, Mónica Barrio, Ismael Fernández, Juan J. Becerra, Pedro G. Herradón, Ana I. Peña, Luis A. Quintela

**Affiliations:** 1Unit of Reproduction and Obstetrics, Department of Animal Pathology, Faculty of Veterinary Medicine, Campus Terra, Universidade de Santiago de Compostela, Avda. Carballo Calero s/n, 27002 Lugo, Spain; uxia.yanez.ramil@usc.es (U.Y.); juanjose.becerra@usc.es (J.J.B.); garcia.herradon@usc.es (P.G.H.); ana.pena@usc.es (A.I.P.); 2Xénese E.T.E., Barreiros, Sedes, 15596 Narón, Spain; monica@xenese-ete.com (M.B.); ismael@xenese-ete.com (I.F.)

**Keywords:** embryo transfer, corpus luteum, ultrasound, rectal palpation, progesterone, heifer

## Abstract

**Simple Summary:**

The selection of recipients plays a key role in embryo transfer (ET) programs. Besides evaluating the animal’s health and reproductive status, this selection is usually performed by assessing luteal function. Rectal palpation (RP), ultrasonography (US), and progesterone determination (P4) are the main techniques used in routine practice. Consequently, we aimed to confirm if US and P4 offer any advantages compared with RP with regard to selecting recipients with higher odds of maintaining pregnancy, and to determine if the CL volume within the ovary (%CLOV) and the presence of a cavity are good predictors of ET outcome. Our results showed that both RP and US are useful techniques to select recipients. Regarding pregnancy rate, we only observed a statistically significant effect for the veterinarian who performed the ET. Therefore, none of the CL measurements studied were good indicators for pregnancy rates after ET procedure.

**Abstract:**

Proper selection of recipients determines the success of embryo transfer (ET) programs. Therefore, the objectives of this study were to assess the accuracy of rectal palpation (RP) in selecting recipients according to the size and firmness of their corpus luteum (CL) compared to ultrasonography (US) and progesterone determination (P4); to check if US or P4 provide additional information to RP with regard to selecting animals with higher odds of maintaining the pregnancy; and to verify the reliability of the presence of a cavity and the volume of the CL within the ovary (%CLOV) as predictors of the ET outcome. In Experiment 1, measurements for the largest and minor diameter (LADCL and MIDCL), CL area, and P4 at ET day were collected, as well as the RP score, in 94 heifers. In Experiment 2, measurements for the LADCL, MIDCL, CL volume, %CLOV, and presence of a cavity were collected, as well as data about the procedure and metabolic markers, in 108 heifers. No differences were found in Experiment 1, whereas in Experiment 2, just a tendency was observed for the variable of veterinarian. Consequently, these results suggest that RP and US are useful methods to select recipients with, that US and P4 do not offer additional data to use to select animals with higher odds of maintaining pregnancy, and that neither %CLOV nor cavitary CL were good indicators for pregnancy rates.

## 1. Introduction

Embryo transfer (ET) is the world’s most widespread technique for rapidly multiplying elite genetics [1]. Although ET practices for cattle have been commercially used for decades, improvements in technique and pregnancy results have remained a topic of research [2,3]. In this regard, it has been stated that despite the many advances that have made ET a routine procedure in many agricultural settings, the process has still not been perfected [4]. 

It is important to note that the success of any ET program is influenced by several factors, among which the proper selection of recipients plays a crucial role [5,6]. Criteria like parity, age, body condition score (BCS), reproductive status, and health status are used to categorize both beef and dairy recipients [1,6]. On the one hand, there are several advantages to using heifers as recipients. The fact that possible reproductive issues derived from previous pregnancies are avoided, along with a small and easy to manipulate uterus, facilitate the achievement of good pregnancy rates [1,6]. However, heifers are more prone to suffer dystocia at calving [1]. On the other hand, the transcervical ET is easier to perform in cows that have already calved, with acceptable pregnancy rates and greater ease of calving. Nevertheless, the fertility of these animals is influenced by other factors such as negative energy balance, high milk production, and postpartum pathologies [7]. In all cases, it is essential to take all possible sanitary measures to avoid any type of embryo loss or abortion and to establish a vaccination program against potential diseases present in the stable or in the area [1].

Under these conditions, having an adequate number of recipients suitable for transfer is a limiting factor for the development of the technique, especially in small farms. It is therefore necessary to maximize the use of these recipients so that they can become pregnant in the shortest possible time. When starting the selection process of recipients, their ability to show estrus within a desired period should be taken into account either by detection of natural estrus or by the use of ovulation synchronization methods for fixed-time ET [8]. Another important key point to consider is the ability of the recipients to ovulate and form a functional corpus luteum (CL) able to produce progesterone for a sufficient time and in a sufficient quantity to act on the progesterone receptors of the uterus, modifying the pattern of uterine secretions necessary to favor the development and maintenance of the embryo [5,9,10,11]. Only during the luteal stage of the estrus cycle does the uterine microenvironment provide an adequate medium for the embryos to develop. As a result, the harmonization of the physiological state of the donor and the recipient in ET procedures lays the foundations for its success [12].

It should be noted that the measurement of plasma progesterone concentration on the day of ET could be used to identify either active or inactive CL. However, this procedure is costly and time-consuming [8]. In farms, the functionality of the CL has been traditionally assessed by rectal palpation (RP), evaluating both the size and the integrity of the CL [13]. In addition, B-mode ultrasonography has also been used for this purpose, considering CL measurements, such as CL area, as indicators of function [14]. More recently, color Doppler ultrasonography has been studied as a novel approach to CL functionality assessment, showing promising results [5,15,16]. However, in daily practice, most veterinarians use either RP or B-mode ultrasonography, as Doppler ultrasonography is not available in most ultrasound equipment yet. 

It has been stated that selecting recipients based on either progesterone levels or CL function is likely to increase the probability of implantation and pregnancy following ET [17]. Therefore, selecting the recipients according to the measurements of the CL may be an efficient strategy to improve ET outcomes. Several characteristics of the CL have been studied in order to determine CL function and to predict pregnancy rates in recipients, such as CL consistency, diameter, area, volume, presence of cavity, echotexture, and blood flow [5,8,11,13,18]. However, controversial results have been reported regarding which of those would be the best indicators or if it is possible to predict pregnancy rates based on CL characteristics.

Moreover, although it has been reported that progesterone production and response to prostaglandin treatments do not differ between cavitary and compact CL [19,20], disparate results have been obtained regarding pregnancy outcome. Some authors described a decreased pregnancy-maintaining ability of cows with a cavitary CL [21], while others considered a cavitary CL as a possible indicator of higher potential of the recipient in maintaining the pregnancy [18].

Consequently, additional research is needed to verify the effect of a cavitary CL on the conservation of pregnancy after ET. Moreover, to our knowledge, the possible influence of the CL volume in relation to total ovarian volume (%CLOV) on the selection of recipients and pregnancy rates has not been evaluated. Therefore, two experiments were carried out to achieve the aims of this study. In Experiment 1, the objective was to assess the accuracy of RP with regard to selecting recipients according to the size and firmness of the CL compared to B-mode ultrasonography (US) and serum progesterone determination (P4); a second objective was to check if US or P4 provide additional information to RP in order to select animals with higher odds of maintaining the pregnancy. In Experiment 2, the objective was to check the reliability of the presence of a cavity and the %CLOV as predictors of the ET outcome.

## 2. Materials and Methods

### 2.1. Animals

In Experiment 1, 94 15–17-month-old Holstein heifers were included in the study. Animals belonged to 7 farms from Galicia (Spain) and were housed in free-stall facilities under intensive management. In Experiment 2, 108 14–18-month-old Holstein heifers were used. Animals belonged to 2 breeding centers from Galicia (Spain) and were housed in free-stall facilities under intensive management. In both experiments, heifers were selected as recipients according to their age (>13 months), health status (negative to IBR, BVD, and *Neospora*), reproductive health, and BCS (2.75–3.5 on a scale of 1–5). The transferred embryos were obtained in vivo from Holstein cows with high genetic value and all of them were scored as Quality 1 according to IETS classification [22]. The non-surgical procedure for ET was used, following the same steps as in regular practice. Embryos were transferred fresh after evaluation, or thawed after cryopreservation, following the adequate protocol (see ET procedure below). In Experiment 1, only frozen embryos using the cryopreservant Ethylene Glycol were used, while in Experiment 2, fresh and frozen embryos with Ethylene Glycol and Glycerol were utilized. Both experiments were conducted in accordance with the European and Spanish Regulations for the protection of animals used for scientific purposes (Directive 2010/63/EU, RD 53/2013).

### 2.2. Synchronization Protocol

In Experiment 1, two different synchronization protocols were used (Figure 1), ensuring that the same protocol was applied on the same farm and set of heifers. The first protocol consisted of the introduction of an intravaginal progesterone release device 1.55 g (PRID^®^ DELTA, Ceva Santé Animale, Libourne, France) and the administration of 50 µg of the GnRH analogue lecirelin (Dalmarelin, Fatro Ibérica, Barcelona, Spain) on Day 0, removal of the progesterone device and administration of 0.15 mg of D-cloprostenol (Dalmazin, Fatro Ibérica, Barcelona, Spain) on Day 7, administration of 50 µg of lecirelin on Day 9, and heat detection thereafter. The second protocol consisted of the introduction of the intravaginal progesterone release device on Day 0, removal of the device and administration of D-cloprostenol on Day 7, and heat detection from Day 9 onwards. The possible influence of these synchronization protocols on the percentage of recipients fit for the ET or on the subsequent pregnancy rates has been previously studied by our group (data not published) and it was observed that both protocols were effective for heat synchronization in heifers.

In Experiment 2, two different synchronization protocols were used as well (Figure 1). The first protocol consisted of the following steps: Day 0, an intravaginal progesterone release device 1.55 g (PRID^®^ DELTA, Ceva Santé Animale, Libourne, France) was introduced; Day 6, 0.15 mg of D-cloprostenol was administered; the progesterone device was removed on Day 7; and heat detection was performed from Day 9 onwards (PRID7PG6). The second protocol was almost identical, the only differences being that the D-cloprostenol was administered on Day 7, and the progesterone device was removed on Day 8 (PRID8PG7).

### 2.3. Recipient Selection

In Experiment 1, heifers were explored by rectal palpation and ultrasonography seven days after the onset of heat. Rectal palpation was performed by two experienced veterinarians with the same selection criteria. The evaluation method was based on the one described by Zemjanis in 1970 [23], which classifies CL according to their size, texture, and firmness (Table 1). Heifers without a luteal structure were classified as “absence of CL”.

Heifers with a hemorrhagic body classified as HB3 were selected and the side of the HB was marked to allow a quicker and easier transfer once the embryo was thawed. Next, an ultrasonography scan of the ovary was performed. A PieMedical Tringa Scan (Esaote, Genoa, Italy), equipped with a 5.0 MHz linear-array transducer, was used. Measurements for the largest diameter of the CL (LADCL), minor diameter of the CL (MIDCL), and CL area (CLA) were obtained.

In Experiment 2, heifers were explored by rectal palpation and ultrasonography on day 7 after the onset of heat. An HS-1500 Scan (Honda Electronics Co., Ltd., Aichi, Japan), equipped with a 7.5 MHz linear-array transducer, was used. Heifers with a CL > 18 mm diameter were included in the study (n = 97). For each selected heifer, a detailed scanning of the ovary was performed, and measurements for LADCL, MIDCL, largest diameter of the ovary (LADOV), and minor diameter of the ovary (MIDOV) were obtained. The volume of the CL (VOLCL) was calculated assuming a perfect sphere, using the formula:$$V=4/3\pi r^{3}$$

When a cavity was present, the volume of the cavity was calculated using the same formula and subtracted from the total volume of the CL. Additionally, the volume of the ovary (VOLOV) was calculated according to the formula:$$V=\frac{4}{3}\pi abc$$
where a, b, and c correspond to height, width, and length of the ovary, respectively. Finally, the percentage of the ovary that corresponds to the CL (%CLOV) was calculated.

### 2.4. Blood Sample Collection and Analysis

Upon completion of the recipients’ examination, blood samples were collected from the coccygeal vein using vacuum tubes without anticoagulants in both experiments. Samples were kept in refrigeration for a maximum of 6 h, and serum was separated after centrifugation at 1500× *g* for 30 min into 0.5 mL aliquots and was stored at −18 °C until analysis.

Serum progesterone (P4) concentrations were measured in both experiments with a commercial progesterone ELISA kit (Progesterone ELISA EIA-1561, DRG Instruments GmbH, Marburg, Germany) following the manufacturer’s instructions. The determinations were performed twice for each sample, and the mean value between both measurements was used. The detection limit for P4 ranged from 0 to 40 ng/mL, with an analytical sensitivity of 0.045 ng/mL. Optical densities were measured in a microplate reader (Multiskan EX, Thermo Fisher Scientific Inc., Waltham, MA, USA).

In Experiment 2, the concentrations of additional biochemical indicators were assessed. All analyses were performed using a digital photometer (Selecta MD200, Barcelona, Spain), except for total protein analysis, for which a portable refractometer was used (J.P. SELECTA S.A., Abrera, Barcelona, Spain). Glucose, total cholesterol, triglyceride, and albumin concentration were determined by a colorimetric endpoint method (BioSystems S.A., Barcelona, Spain). Hepatic enzymes alanine aminotransferase (ALAT), aspartate aminotransferase (ASAT), and gamma-glutamyltransferase (GGT) were determined by using BioSystems reagents. Urea was also analyzed through a colorimetric enzymatic method with Spinreact reagents (Spinreact, S.A.U., Sant Esteve de Bas, Spain). Non-esterified fatty acids (NEFA) and β-hydroxybutyrate acid (BHBA) were determined by kinetic enzymatic kits (Randox Laboratories Ltd., Antrim, UK).

### 2.5. Embryo Transfer Procedure

Once the recipients were selected, they were prepared for the transfer following the non-surgical procedure in both experiments. The ET was performed by one experienced veterinarian in Experiment 1, while in Experiment 2, the procedure was carried out by 2 different, experienced veterinarians. First, animals were sedated using 6–10 mg xylazine (Rompun 20 mg/mL, Bayer Animal Health GmbH, Leverkusen, Germany). Then, epidural administration of 80 mg of procaine (Pronestesic 40 mg/mL, Fatro Iberica, Barcelona, Spain) was performed to immobilize cows as much as possible to avoid injuries in the endometrium during the transfer. Additionally, in Experiment 2, 500 mg of the anti-inflammatory agent flunixin meglumine (Finadyne 50 mg/mL, Merck Sharp & Dohme Animal Health, S.L., Kenilworth, NJ, USA) was administered intramuscularly (n = 9) to avoid the release of endogen prostaglandins as a consequence of the transfer. Finally, the entire perivulvar area was wiped with paper to avoid introducing contamination into the vagina and uterus.

After embryo collection/thawing (Appendix A), the straw was placed into a Cassou rod of 3 mm diameter (Minitub Iberica S.L., Tarragona, Spain). The rod was covered with a sterile sheath with steel tip and side holes, and it was protected with a plastic sanitary sleeve (Chemise Sanitaire, IMV Technologies, L’Aigle, France). Once the rod was introduced in the cervix, the sanitary sleeve was retracted and the rod was passed into the uterus halfway into the uterine horn ipsilateral to the ovary with the CL, where the embryo was smoothly placed. Pregnancy diagnoses were performed at 30–40 days after embryo transfer using ultrasonography, and recipient cows were classified as pregnant or not pregnant.

### 2.6. Statistical Analysis

In Experiment 1, serum P4 levels, LADCL, MIDCL, and CLA were considered continuous variables, and the rectal palpation classification of the CL (RPCL) was considered a categorical variable, classified as shown in Table 1. First, a Pearson’s correlation analysis was performed including serum P4 levels, LADCL, MIDCL, and CLA. To compare RPCL, a Oneway ANOVA test was performed using the general linear model (GLM) tool, including serum P4, LADCL, MIDCL, and CLA as dependent variables, and the RPCL as factor. The HSD Tukey test was used to check the significant differences between ovarian structures, and homogeneity of variances was checked using Levene’s test.

Additionally, the influence of rectal palpation, ultrasonography, and progesterone measurements on the ET outcome was tested. A Oneway ANOVA test was performed using the GLM tool, including pregnancy as independent variable and serum P4, LADCL, MIDCL, and CLA as dependent variables.

In Experiment 2, the following were considered as continuous variables: LADCL, MIDCL, LADOV, MIDOV, VOLCL, %CLOV, serum P4, BCS, glucose, total cholesterol, triglyceride, albumin, total protein, ALAT, ASAT, GGT, urea, BHB, and NEFA. As for the categorical variables, they were considered as follows: pregnant (YES/NO), CL cavity (YES/NO), flunixin meglumine administration (YES/NO), type of embryo (Ethylene glycol, glycerol, fresh), veterinary who performed the transfer (n°1/n°2), and synchronization protocol (PRID7PG6/PRID8PG7). We determined the relative contribution of each factor to the probability of pregnancy using a backward stepwise binary logistic regression. We included the pregnancy outcome as dependent variable and the remaining variables as independent factors.

All analyses were conducted in SPSS version 28.0 for Windows (SPSS Inc., Chicago, IL, USA). Differences were considered significant at *p* ≤ 0.05.

## 3. Results

### 3.1. Experiment 1

Of the 94 recipients evaluated for the ET, 51 were selected according to the rectal palpation criteria, i.e., presence of a HB3 in the ovaries. Data from the 94 animals were used to assess the correlation between the different techniques used to evaluate the CL. On the other hand, of the 51 animals selected for the transfer, 2 heifers were excluded due to data loss. Therefore, data from 49 selected heifers were used to test the influence of the measurements obtained on the ET outcome. The pregnancy rate in this experiment was 44.9%.

Results for the Pearson’s correlation analysis showed that there is a positive, statistically significant correlation between serum P4 concentration, CLA, and MIDCL (Table 2).

Regarding RPCL, no observations were made for HB1 and CL1, i.e., none of the animals included in the study presented any of these structures in their ovaries. Results for the Oneway ANOVA are displayed in Table 3. 

Concerning the influence on pregnancy rates, the Oneway ANOVA test showed no statistically significant differences between pregnant and non-pregnant heifers regarding the variables assessed (Table 4).

### 3.2. Experiment 2

Of the 108 heifers initially evaluated, 97 were finally included in the study as recipients. Of these 97 heifers, 3 had two CL, and were excluded from the study. Consequently, the total number of animals included was 94, of which 67 animals were pregnant after ET, and 27 remained non-pregnant (pregnancy rate = 69%).

Results for the binary logistic regression showed that there are not significant differences between pregnant and non-pregnant heifers concerning the variables included in the model, except for a tendency observed for the veterinarian who performed the ET (OR = 2.476, *p =* 0.06; Table 5 and Table 6). The Hosmer and Lemeshow test was not significant (*p =* 0.799), and the overall significance of the model was *p* = 0.05.

## 4. Discussion

Our results for the correlation analysis in Experiment 1 showed that there is a positive, statistically significant correlation between serum P4 concentrations and CL measurements like CLA and MIDCL. Consequently, B-mode US may provide proper information about CL function and may be a useful tool to select recipients before ET. These findings are in accordance with those observed by other researchers, who stated that CLA is closely related to P4 concentrations and has the greatest association with luteal function during CL development [24,25]. In this regard, a recent study by our laboratory [16] showed that CLA could be the parameter of choice when using B-mode US to assess luteal function. However, it should be noted that Doppler US is becoming more popular nowadays, and numerous investigations are carried out to assess its accuracy as a tool for reproductive management [26,27,28,29]. So far, research has been made on the usefulness of Doppler US to determine CL function, and promising results were obtained, especially regarding luteolysis detection and early pregnancy diagnosis [16,30,31,32,33]. With respect to ET, the relationship between CL blood flow and recipient selection, pregnancy rates, and early pregnancy diagnosis has been studied, observing more favorable results in the first case [5,15,34].

Regarding RP, results for the Oneway ANOVA analysis showed that there were statistically significant differences between HB3 and HB2 for CLA, LADCL, and MIDCL. Additionally, differences were also observed between HB3 and CL2 for CLA and MIDCL. Finally, serum P4 concentrations significantly differ between HB3 and the absence of CL. On the other hand, no significant differences were found between HB3 and CL3. Due to the similar values obtained for HB3 and CL3 with respect to size measurements and P4 concentration, US and P4 determination do not seem to be proper approaches to discern between these two luteal structures, which are present from days 6 to 17 of the estrous cycle. To correctly perform ET, recipients should ideally be around day 7 of the estrous cycle, according to the age of the embryo [35]. Our results suggest that US and P4 determination are useful for identifying developing (HB1, HB2) or under-regressed CL (CL2, CL1), but that only RP would allow us to distinguish between HB3 and CL3, using the firmness as an indicator of luteal age. As the estrous cycle progresses, the CL becomes firmer and more compact. However, there is not a significant correlation, neither between its firmness and progesterone concentration at a mature state, nor between its firmness and luteal size [24]. This characteristic might be evaluated using US considering the echotexture [36]. Nevertheless, although it has been stated that luteal tissue heterogeneity is considered a potential indicator of CL function [14], no significant differences in CL echotexture characteristics were found between days 7 and 18 of the estrous cycle [37].

As for pregnancy rates, no statistically significant differences were observed for serum P4 concentration, CLA, LADCL, and MIDCL in cows selected via RP, i.e., those who had a HB3 present in the ovary at the time of ET. These findings suggest that neither P4 determination nor US provide additional information to RP that could lead to the selection of recipients with higher odds of maintaining the pregnancy. However, it should be considered that there are additional factors that might condition the outcome of the ET and might be used as possible indicators of pregnancy rate. In this regard, Frade et al. [38] reported that the manifestation of estrous behavior may have a positive relationship with pregnancy rates in ET programs, which may be due to the effects of the estrogens on the endometrial tissue. Similarly, other researchers also stated that cows that displayed estrus had better ovarian responses (larger follicles, larger CL, and greater P4 concentrations) and improved pregnancy outcomes after timed artificial insemination programs [39].

Concerning Experiment 2, our results for the binary logistic regression showed that, among all the variables studied, only the veterinarian who performed the ET tended to influence the subsequent pregnancy rate. It is widely accepted that the practitioner competence is a key point in the success of ET, along with good-quality embryos and recipient management [2]. It should be mentioned that both veterinarians who performed the ET in this experiment had significant experience in this field and pregnancy rates were optimal (81.6 and 61.7%); nevertheless, these results emphasize the importance of this factor and disclose the diversity of aptitudes even among qualified professionals.

On the other hand, none of the CL measurements showed significant differences between pregnant and non-pregnant recipients. As in Experiment 1, LADCL and MIDCL did not differ between groups. Similarly, neither the VOLCL, nor the %CLOV, seemed to be reliable indicators of the ET outcome. As mentioned above, the presence of a functional CL in the ovary is indispensable in order to maximize the odds of pregnancy after ET. Recipients in this experiment were selected according to the size of the CL (LADCL > 18 mm). Therefore, the lack of significance can be explained by the fact that only heifers with a functional CL were included in the study, i.e., once CL size is beyond a certain threshold, higher measurements do not imply better odds of pregnancy. Similar effects have been previously described, as researchers claimed that, in ET procedures, there is a positive relationship between pregnancy rate and CL area until luteal tissue reaches a certain size [5].

In terms of cavitary CL, our results showed that the pregnancy rate did not differ between cows with cavitary or homogeneous CL, as previously reported by Perez-Marin [40]. It is known that the presence of a cavity inside the CL is not a detrimental factor for progesterone production. However, different results have been reported about this topic. Some researchers stated that the presence of a luteal cavity did not influence progesterone production throughout the complete estrous cycle [40], while others described higher progesterone levels, and higher pregnancy rates, in heifers with a cavitary CL [11,18]. Consequently, it is clear that further research is needed to shed light on this matter and clarify if the luteal cavity may influence the outcome of ET and artificial insemination (AI).

Regarding the type of preservation used for the embryos, we observed higher pregnancy rates in the fresh group. In this regard, Hansen [4] also described lower (7.4%) pregnancy rates for cows receiving cryopreserved embryos than those receiving fresh embryos. However, no statistically significant differences were present in our study with respect to ethylene glycol and glycerol groups for pregnancy rate. One possible explanation could be the lack of statistical power due to the low number of heifers in the fresh and glycerol groups. 

Another aspect to consider is the metabolic status and health of the recipients. We did not observe statistically significant differences regarding BCS or the metabolic indicators analyzed between pregnant and non-pregnant recipients. As animal husbandry and management factors, including disease status, nutrition, and reproductive health, play a crucial role in the success of the ET, the selection of the recipients must include the validation of these indicators [2,6]. In our study, a previous examination of heifers was performed to select which animals would be enrolled. Therefore, it is very probable that, as only heifers with adequate health were selected, the detrimental effects that nutritional issues may have on pregnancy rates were screened at this point.

Finally, no statistically significant differences were observed for the synchronization protocol and the administration of flunixin meglumine. Concerning the synchronization protocol, it should be considered that there is a wide variety of protocols used for ET in the literature. In this way, protocols including progesterone release devices, like the ones in our study, work well to synchronize heat in heifers [41]. Therefore, as long as heat detection was properly performed and ET was carried out in due time, it was expected to obtain similar results regarding pregnancy rates. However, the difference of sample size between groups cannot be ignored, as it may impair the statistical power of the model. As for the use of flunixin meglumine, it has been reported that its administration at the time of ET improved pregnancy rates in cows [42]. Similarly, Purcell et al. reported a beneficial effect of this treatment on pregnancy rates, but this was dependent of the location effect [43]. Additionally, it has been observed that the administration of flunixin meglumine to cows with subclinical endometritis has a positive effect on pregnancy rate [44]. Given the disparity of the results, additional research including a higher number of animals is needed.

## 5. Conclusions

In conclusion, RP and US are useful for performing the selection of recipients for ET by means of luteal function evaluation. Moreover, RP would allow us to discern between different luteal structures (HB3 and CL3) that are present around the days of ET and that cannot be distinguished by US or P4 determination. Additionally, in recipients selected using RP, measurements for luteal area, diameters, and P4 determination do not seem to offer additional data that would refine the selection of animals with higher odds of maintaining the pregnancy. Finally, neither %CLOV nor the presence of a luteal cavity appear to be good indicators for pregnancy rates in ET procedures. Therefore, further research is required to enhance the selection of recipients and additional studies on innovative approaches like Doppler ultrasonography and uterine features should be performed.

## Figures and Tables

**Figure 1 animals-13-02865-f001:**
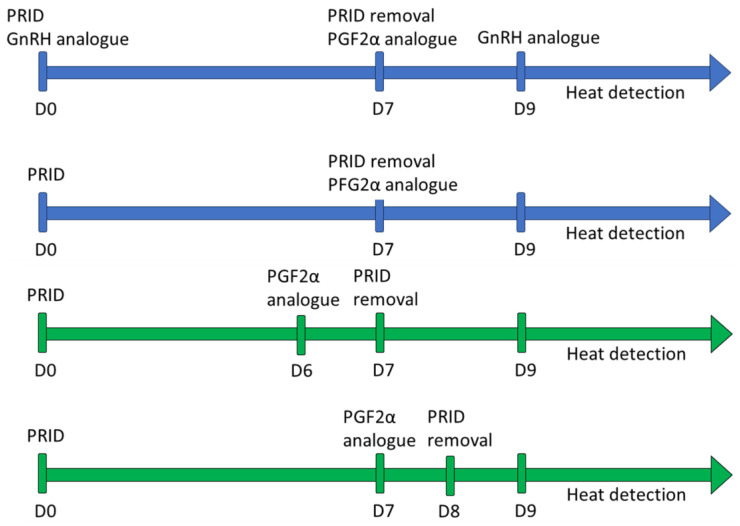
Schemes of the different synchronization protocols followed in Experiment 1 (blue) and Experiment 2 (green) to synchronize 94 and 108 Holstein heifers, respectively, as recipients for embryo transfer.

**Table 1 animals-13-02865-t001:** Corpus luteum (CL) classification followed to select 94 Holstein heifers as recipients for embryo transfer procedure. Rectal palpation was performed seven days after the onset of heat, following a heat synchronization protocol (Adapted from Zemjanis, 1970 [23]).

CL Classification	Days of Cycle	Description
Absence of corpus luteum	1–2	Collapsed follicle
Hemorrhagic body 1 (HB1)	2–3	Soft, friable, developing CL, <1 cm
Hemorrhagic body 2 (HB2)	3–5	Soft CL, 1–2 cm
Hemorrhagic body 3 (HB3)	6–7	Consistent CL, >2 cm, still mobile, with palpable crown
Corpus luteum 3 (CL3)	8–17	Big and firm CL, crown integrated into the ovary
Corpus luteum 2 (CL2)	18–20	Hard CL, under regression
Corpus luteum 1 (CL1)	20–21	Very hard CL, <1 cm

**Table 2 animals-13-02865-t002:** Results for the Pearson’s correlation analysis, including 94 Holstein heifers, for the variables serum progesterone concentration (serum P4), corpus luteum area (CLA), largest diameter of the CL (LADCL), and minor diameter of the CL (MIDCL).

	Serum P4	CLA	LADCL	MIDCL
Serum P4	1	0.335 **	0.189	0.368 **
CLA	-	1	0.758 **	0.799 **
LADCL	-	-	1	0.511 **
MIDCL	-	-	-	1

** *p* ≤ 0.01.

**Table 3 animals-13-02865-t003:** Results for the Oneway ANOVA test, including 94 Holstein heifers, for the variables serum progesterone concentration (Serum P4), corpus luteum area (CLA), largest diameter of the CL (LADCL), and minor diameter of the CL (MIDCL) according to the classification of the luteal structure present in the ovary, evaluated via rectal palpation (Zemjanis, 1970 [23]): hemorrhagic body 2 (HB2), hemorrhagic body 3 (HB3), corpus luteum 3 (CL3), and corpus luteum 2 (CL2).

	n	Serum P4 (ng/mL)	CLA (cm^2^)	LADCL (cm)	MIDCL (cm)
Absent CL	7	2.17 ± 1.99 ^a^	-	-	-
HB2	14	7.67 ± 5.70 ^ab^	3.04 ± 1.10 ^a^	2.39 ± 0.46 ^a^	1.53 ± 0.29 ^a^
HB3	51	7.81 ± 4.37 ^b^	4.16 ± 1.02 ^b^	2.81 ± 0.43 ^b^	1.90 ± 0.32 ^b^
CL3	15	7.52 ± 6.14 ^ab^	3.49 ± 1.01 ^ab^	2.66 ± 0.67 ^ab^	1.72 ± 0.36 ^ab^
CL2	7	3.84 ± 2.98 ^ab^	2.54 ± 1.03 ^a^	2.27 ± 0.51 ^ab^	1.35 ± 0.29 ^a^

^a,b^ Different letters indicate statistically significant differences in the same column.

**Table 4 animals-13-02865-t004:** Results for the Oneway ANOVA test, including 49 Holstein heifers, for the variables serum progesterone concentration (Serum P4), corpus luteum area (CLA), largest diameter of the CL (LADCL), and minor diameter of the CL (MIDCL) according to the pregnancy diagnosis 30–40 days after the embryo transfer procedure: negative (NO) or positive (YES).

Pregnancy	n	Serum P4 (ng/mL)	CLA (cm^2^)	LADCL (cm)	MIDCL (cm)
NO	27	9.02 ± 8.31	3.92 ± 1.19	2.74 ± 0.42	1.87 ± 0.35
YES	22	9.54 ± 7.16	4.36 ± 1.00	2.93 ± 0.48	1.92 ± 0.33

**Table 5 animals-13-02865-t005:** Descriptive statistics (mean ± standard deviation) of Experiment 2 for the largest diameter of the corpus luteum (LADCL), minor diameter of the CL (MIDCL), volume of the CL, the percentage of ovary that corresponds to the CL (%CLOV), serum progesterone concentration (Serum P4), body condition score (BCS), glucose, total cholesterol, albumin, total protein, triglyceride, ALAT, ASAT, GGT, urea, BHB, and NEFA of 94 heifers according to their pregnancy diagnosis after embryo transfer.

Variable	Pregnant (n = 67)	Not Pregnant (n = 27)
LADCL (mm)	28.55 ± 3.73	27.86 ± 3.95
MIDCL (mm)	18.96 ± 3.93	19.60 ± 3.99
VOLCL (mm^3^)	7148.52 ± 3064.74	7165.99 ± 2876.72
%CLOV	48.11 ± 15.71	45.62 ± 11.13
Serum P4 (ng/mL)	16.30 ± 11.89	17.45 ± 13.28
BCS	2.94 ± 0.27	3.03 ± 0.31
Glucose (mg/dL)	76.85 ± 28.32	76.13 ± 16.35
Total cholesterol (mg/dL)	118.29 ± 43.12	112.22 ± 31.00
Albumin (g/L)	31.82 ± 7.85	30.75 ± 7.34
Total proteins (g/L)	62.12 ± 4.27	63.00 ± 7.29
Triglycerides (mg/dL)	54.39 ± 76.03	62.48 ± 87.56
AST (U/L)	116.99 ± 63.54	119.48 ± 60.99
ALT (U/L)	51.92 ± 38.13	49.77 ± 53.97
GGT (U/L)	25.73 ± 7.39	27.21 ± 11.48
Urea (mg/dL)	31.61 ± 10.41	30.21 ± 9.36
BHB (mmol/L)	0.36 ± 0.32	0.32 ± 0.21
NEFA (mmol/L)	0.58 ± 0.81	0.79 ± 1.13

**Table 6 animals-13-02865-t006:** Descriptive statistics of Experiment 2 for the percentage of pregnant Holstein heifers (n = 94) after embryo transfer regarding the following variables: presence of a corpus luteum (CL) with a cavity, administration of flunixin meglumine at embryo transfer, type of cryoprotectant agent used for embryo preservation, veterinarian who performed the embryo transfer, and the synchronization protocol used.

Variable		n	Pregnant Heifers
CL with cavity	YES	33	22/33 (66.7%)
	NO	61	45/61 (73.4%)
Flunixin meglumine	YES	9	5/9 (55.6%)
	NO	85	62/85 (72.9%)
Embryo preservation	Ethylene glycol	64	44/64 (68.7%)
	Glycerol	17	12/17 (70.6%)
	Fresh	13	11/13 (84.6%)
Veterinarian *	1	48	39/48 (81.2%)
	2	45	27/45 (61.7%)
Synchronization protocol	PRID7PG6	76	55/76 (72.2%)
	PRID8PG7	18	12/18 (66.7%)

* *p* = 0.06.

## Data Availability

Data generated and/or analyzed during the current study are available from the corresponding author on reasonable request.

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
