# Peer review of "Assessment of Luteal Function Using Rectal Palpation, B-Mode Ultrasonography, and Progesterone Determination to Improve Recipient Selection in Embryo Transfer Programs"

_animals, 2023, doi:10.3390/ani13182865_

Round 1

Reviewer 1 Report

This paper presents interesting results but is quite difficult to follow and there are confounding factors that render some of the statistics less powerful or erroneous.  For example, the statistical analysis of Experiment 1 does not mention the different estrus synchronization protocols.

L 19 replace “check” with ‘determine’

L 20 and 21 RP with US? US with P4? This becomes clear later in the manuscript, but as written it is confusing.

L 22 statistically significant effect of veterinarian

What did the veterinarians do differently?  Did they report their evaluations so the authors might determine what were significant factors for each vet?

L 35 what were the metabolic markers

L 49 and 58 ‘conditioned’ should be replaced by ‘influenced’

L 54 How does a small uterus facilitate pregnancy rates?

L 59 how does milk production influence fertility?

L 130 this is not clear. Did each farm use a different synchronization protocol? Was farm location considered a statistical variable?

How was the effect of freezing and thawing protocols and comparison to fresh embryos analyzed?

How did the authors determine which cows, in which pre-transfer treatment groups, received fresh or differently-frozen embryos?

L 133 define lecirelin when it is first mentioned

L 150 the use of 2 different synchronization protocols in each experiment, with no repeated protocols across experiments, confounds the entire experiment and makes statistical analysis difficult.  Is there an explanation for using 4 different protocols?

L 210 do the authors believe there was no difference in the skill of the 3 vets performing the ETs? They did have slightly different pregnancy rates.

L 216 define/explain flunixin meglumine

L 218 this difference of giving prostaglandin after ET in experiment 2 and not in experiment 1 may have confounded this study.  If this was part of the experimental design, it should be explained.

L 221 this section is difficult to understand.  The embryo transfer is described first, then the narrative jumps to the cryopreservation and thawing protocols.  “Once the collection was finished” seems out of place.  Assuming the collection means the collection of embryos from another, presumably superovulated donor, there is no information about the donors, the collection procedure, or cryopreservation procedures.

L 222 this is the first mention that the embryos were frozen

L 281 what parameters of the rectal palpation were used to select the 51 recipients?

L 286 was the pregnancy rate calculated regardless of embryo state (fresh, frozen method 1 and frozen method 2)?  Is 44.9% considered an acceptable pregnancy rate?

L 294 please explain why there were no observatons for HB1 and CL1.  The data were not collected or they were not analyzed?

Table 4 the first column should be labeled pregnancy rather than gestation

L 315 what may have resulted in the differences in pregnancy rates in experiments 1 and 2?

These pregnancy rates don’t seem to reflect the overall pregnancy rates reported (for example, 44.9% and 69% for experiments 1 and 2, respectively).

L 410 is the 7.4% pregnancy rate the difference between fresh and frozen embryos?  Clarify.

There are a few minor English corrections needed.

Author Response

Reviewer 1:

This paper presents interesting results but is quite difficult to follow and there are confounding factors that render some of the statistics less powerful or erroneous.  For example, the statistical analysis of Experiment 1 does not mention the different estrus synchronization protocols.

 L 19 replace “check” with ‘determine’

The manuscript was modified according to the reviewer’s suggestions.

 L 20 and 21 RP with US? US with P4? This becomes clear later in the manuscript, but as written it is confusing.

The sentence was modified to avoid confusion.

L 22 statistically significant effect of veterinarian

The manuscript was modified according to the reviewer’s suggestions.

What did the veterinarians do differently?  Did they report their evaluations so the authors might determine what were significant factors for each vet?

The protocol is the same for all the veterinarians; only the skill of each one in inserting the catheter in the right place, as quickly and with as little damage as possible, differs. The veterinarians' impressions about their performance were not recorded.

L 35 what were the metabolic markers

The metabolic markers analyzed were: glucose, total cholesterol, triglycerides, albumin, hepatic enzymes (ALAT, ASAT and GGT), urea, NEFAs and BHBA. This information was included in section 2.4 and not in the abstract due to the limit of words.

L 49 and 58 ‘conditioned’ should be replaced by ‘influenced’

The manuscript was modified according to the reviewer’s suggestions.

L 54 How does a small uterus facilitate pregnancy rates?

We meant to say that a small uterus would be easy to manipulate, compared to those of cows who have already calved, when performing the embryo transfer (with the exception of passing the rod through the cervix, as it is mentioned later in the paragraph), so it would be helpful to achieve good pregnancy rates, as a good technique plays a crucial role.

L 59 how does milk production influence fertility?

In lactating cows, milk production would prevail over reproductive function. Therefore, cows under negative energy balance would not meet the requirements to recover their reproductive health, including resumption of ovarian cyclicity, uterine involution and hormonal balance. To make this clearer in the manuscript, we remarked that we refer to high milk yields.

L 130 this is not clear. Did each farm use a different synchronization protocol? Was farm location considered a statistical variable?

With the aim of maintaining, to the best extent possible, the regular operation of the farms, the protocols used were chosen based on those routinely employed in these settings. Since we have previously verified that the protocols utilized do not have a significant impact on the selection of the recipients or the pregnancy rate, as mentioned in the manuscript,  farm location has not been included as a variable in the statistical analysis.

How was the effect of freezing and thawing protocols and comparison to fresh embryos analyzed?

Each protocol was consistently performed in the same manner, as described in the materials and methods section. For frozen embryos, the freezing process was unknown since they were purchased embryos. The analysis was conducted by including the variable in the logistic regression analysis.

How did the authors determine which cows, in which pre-transfer treatment groups, received fresh or differently-frozen embryos?

The choice of embryo type was the responsibility of the livestock farmer, guided by the veterinarian, based on genetic merit and availability considerations. All of this was always part of the veterinarians' daily work.

L 133 define lecirelin when it is first mentioned

The definition of lecirelin (GnRH analogue) was included in the manuscript as suggested.

L 150 the use of 2 different synchronization protocols in each experiment, with no repeated protocols across experiments, confounds the entire experiment and makes statistical analysis difficult.  Is there an explanation for using 4 different protocols?

Firstly, each experiment was conducted in different years, so the protocols might vary based on the knowledge available at the time of each experiment. Within each experiment, the protocols that were commonly being used to induce estrus on each farm were employed, aiming to prevent any confusion during implementation.

L 210 do the authors believe there was no difference in the skill of the 3 vets performing the ETs? They did have slightly different pregnancy rates.

All the veterinarians had several years of experience, although it is true that one of them had more years of experience than the others, which could provide more expertise when it came to embryo transfer. Although this difference is already apparent in the results of the statistical analysis, it persists when we analyze farm by farm. Based on the results, we have to consider that indeed, Veterinarian 1 exhibits greater skill than Veterinarian 2.

L 216 define/explain flunixin meglumine

The definition of flunixin meglumine (anti – inflammatory agent) was included in the manuscript as suggested.

L 218 this difference of giving prostaglandin after ET in experiment 2 and not in experiment 1 may have confounded this study.  If this was part of the experimental design, it should be explained.

As previously mentioned, the experiments have been conducted in different years, which is why there are differences resulting from improvements that have been gradually introduced over time. While it's true that this could pose a confounding factor between Experiment 1 and Experiment 2, since both experiments are treated independently, we do not believe it is relevant to the final conclusion.

L 221 this section is difficult to understand.  The embryo transfer is described first, then the narrative jumps to the cryopreservation and thawing protocols.  “Once the collection was finished” seems out of place.  Assuming the collection means the collection of embryos from another, presumably superovulated donor, there is no information about the donors, the collection procedure, or cryopreservation procedures.

The sentence “Once the collection was finished” was modified according to the reviewer’s suggestions. Regarding thawing protocols, it has been suggested that this part may be removed from the manuscript and put as supplementary materials. Therefore, we will wait until the next revision to know all reviewer’s and editor’s opinion about it and make the modifications, if appropriate.

L 222 this is the first mention that the embryos were frozen

A mention that embryos were transferred fresh or thawed was included in section 2.1.

L 281 what parameters of the rectal palpation were used to select the 51 recipients?

The information was included in the manuscript, in the first paragraph of section 3.1., according to the reviewer’s suggestions.

L 286 was the pregnancy rate calculated regardless of embryo state (fresh, frozen method 1 and frozen method 2)?  Is 44.9% considered an acceptable pregnancy rate?

In Experiment 1, all transferred embryos were frozen with ethylene glycol. In Experiment 2, the different methods were included in the logistic regression.

According to the reference Hansen (2020), pregnancy rates after embryo transfer, including both fresh and frozen embryos, ranged from 34.4 to 55.3 %. Therefore, we used the term “acceptable”, as the pregnancy rate observed in our study fits within the mentioned range.

L 294 please explain why there were no observatons for HB1 and CL1.  The data were not collected or they were not analyzed?

None of the animals included in the study presented any of these structures in their ovaries. This information was included in the text after Table 2.

Table 4 the first column should be labeled pregnancy rather than gestation

The table was modified according to the reviewer’s suggestions.

L 315 what may have resulted in the differences in pregnancy rates in experiments 1 and 2? These pregnancy rates don’t seem to reflect the overall pregnancy rates reported (for example, 44.9% and 69% for experiments 1 and 2, respectively).

In Experiment 1, 7 commercial farms were used, whereas in Experiment 2, animals from two official centers dedicated to Embryo Transfer (TE) were utilized. These centers were directly supervised by veterinarians in both cases, and the animals were solely dedicated to this purpose. The results from Experiment 1 resemble those generally obtained in other studies, while those from Experiment 2 are superior due to the more meticulous handling of the animals.

L 410 is the 7.4% pregnancy rate the difference between fresh and frozen embryos?  Clarify.

7.4% was the difference reported in the referenced study for pregnancy rate between fresh and frozen embryos. In the present manuscript, pregnancy rates were 68.7% and 70.6% for frozen embryos, and 84.6% for fresh embryos.

Reviewer 2 Report

Line 105 I would like know why you don’t have merge the results of your observations done by manual palpation, ultrasonography and P4 in the two experiments ?

Line 157 : what do you think about the subective evaluation of the firmness of a corpus luteum ? Indeed, you write in your discussion that « is not a significant correlation neither between its firmness and progesterone concentration at a mature state, nor between its firmness and luteal size » (Reference 23)

Line 160 Table 1 What means « crown integrated into the ovary » ? Can we consider that in such case the crown (also described like a champagne cork) is not palpable at the surface of ovary ?

Line 208 : the number of embryo from each category (fresh, ethylene glycol or glycerol) could be described in experiment 1

Line 218 “were administered intramuscular (n = 9) “ : why an higher number of  heifers don’t have been treated with flunixine  ¿

Line 281 : it could be interesting to explain why 41 (43-2) heifers don’t have been selected for an embryo transfer. This number of selected animals is quite different in experiment 1 and 2. Can we see an indirect effect of the synchronisation protocol ?  

Line 313 97 cows or heifers  ¿

Line 315 : how explain such differences in pregnancy rate between the two experiments : 44,8 vs 69 % ?

Line 321 have you evaluated the pregnancy rate according to the diameter of corpus luteum < 2cm (if any) ? May be in the first experiment, the number of such heifers has been higher than in the second experiment.

Line 389 such absence of difference could be normal according to the high value of the cutt off of the different measures done for the corpus luteum (> 18 mm for the diameter).  That’s what you written in lines 394 to lines 397. I am still convinced that such measurements is better by US than by RP.

Line 425 a positive effect of flunixine has been observed in beef cow recipients after subclinical endometritis (see Barnes M. et al. Effect of subclinical endometritis and flunixin meglumine administration on pregnancy in embryo recipient beef cows. Theriogenology 201 (2023) 76-82)

Your experiment add  some information on the best propedeutic method used for the selection of recipient. I have been a little bit stonish by your conclusion in favor of RP.

Attending your answers to the following questions, I would recommand to suspend the publication of this paper

·         I would like know also why you don’t have merge the data collected by RP, US and P4 in your two experimental group.

·         Moreover, coud you explain why you don’t have compare the sensibility, specificity, positive and negative predictive value of RP and US diagnosis compared to P4 concentration.

·         Could you explain also the utility to study the possible effect of the two synchronisation protocol ?  

Author Response

Line 105 I would like know why you don’t have merge the results of your observations done by manual palpation, ultrasonography and P4 in the two experiments?

The experiments were conducted in different years, different locations, with different animals and protocols, which we fear would increase variability.

Line 157 : what do you think about the subective evaluation of the firmness of a corpus luteum ? Indeed, you write in your discussion that « is not a significant correlation neither between its firmness and progesterone concentration at a mature state, nor between its firmness and luteal size » (Reference 23)

Subjective evaluation of the firmness of a CL can be useful in certain aspects of daily practice. For example, it may be a good indicator of the moment of the estrous cycle, as it goes from a soft texture during the first 5 days to a hard texture during luteal regression, which is useful to decide if that CL can be sensitive to the action of PGF2α or if it is the right moment to perform the ET. However, we agree that this is a subjective approach, and its precision may be not perfectly accurate, as during the mature state the firmness does not vary substantially.

Line 160 Table 1 What means « crown integrated into the ovary »? Can we consider that in such case the crown (also described like a champagne cork) is not palpable at the surface of ovary?

Instead of the typical champagne cork shape, it is more like a palpable button, which makes it harder to palpate and estimate its size.

Line 208: the number of embryo from each category (fresh, ethylene glycol or glycerol) could be described in experiment 1

In Experiment 1, all embryos were frozen with Ethylene glycol. An explanation about this so it would be clearer was included in section 2.1.

Line 218 “were administered intramuscular (n = 9) “ : why an higher number of  heifers don’t have been treated with flunixine  ¿

As the experiments were carried out in commercial farms/centers, we tried to adapt as much as possible to their normal operation.

Line 281 : it could be interesting to explain why 41 (43-2) heifers don’t have been selected for an embryo transfer. This number of selected animals is quite different in experiment 1 and 2. Can we see an indirect effect of the synchronisation protocol ? 

Those 43 non – selected heifers were excluded due to the presence of a luteal structure different from a HB3 in the ovaries. The selection criteria differ between experiments, as well as the location and the animals. Yes, there could indeed be an indirect effect of the protocol, although the differences are small, and it's likely that the most significant cause is the greater variability in the animals between the two experiments.

Line 313 97 cows or heifers  ¿

The manuscript was modified according to the reviewer’s suggestions.

Line 315 : how explain such differences in pregnancy rate between the two experiments : 44,8 vs 69 % ?

In Experiment 1, 7 commercial farms were used, whereas in Experiment 2, animals from two official centers dedicated to Embryo Transfer (TE) were utilized. These centers were directly supervised by veterinarians in both cases, and the animals were solely dedicated to this purpose. The results from Experiment 1 resemble those generally obtained in other studies, while those from Experiment 2 are superior due to the more meticulous handling of the animals.

Line 321 have you evaluated the pregnancy rate according to the diameter of corpus luteum < 2cm (if any) ? May be in the first experiment, the number of such heifers has been higher than in the second experiment.

The evaluation of pregnancy rate according to a diameter of CL <2cm was not possible, as heifers were first selected by RP and only those who met the criteria were used for the embryo transfer, with CL >2cm (the mean value for the largest diameter of the CL is displayed in Table 3.

Line 389 such absence of difference could be normal according to the high value of the cutt off of the different measures done for the corpus luteum (> 18 mm for the diameter).  That’s what you written in lines 394 to lines 397. I am still convinced that such measurements is better by US than by RP.

We agree with the reviewer that US is a more accurate approach to exactly determine the size of the CL. The size determined by RP can only be subjective, even with several years of experience. We chose that cut off following the references mentioned, but of course a different choice could have led to different results. Nevertheless, we want to make clear that we are not saying that RP is better than US in ET transfer programs. Both techniques are useful and work with acceptable results, as it has been stated in the conclusions.

Line 425 a positive effect of flunixine has been observed in beef cow recipients after subclinical endometritis (see Barnes M. et al. Effect of subclinical endometritis and flunixin meglumine administration on pregnancy in embryo recipient beef cows. Theriogenology 201 (2023) 76-82)

The manuscript was modified according to the reviewer’s suggestions.

Your experiment add  some information on the best propedeutic method used for the selection of recipient. I have been a little bit stonish by your conclusion in favor of RP.

Concerning this comment, we would like to make clear that we are not in favor of RP, meaning that “it is better that US”. We observed that both techniques are useful to select recipients for ET, and the firmness determined by RP would allow to differentiate between HB3 and CL3. However, that differentiation does not mean that the outcome of the ET would be substantially better if the heifers are selected by RP. Indeed, we wanted to determine the opposite, if US would significantly improve pregnancy rates compared to RP, but our results showed no differences.

Attending your answers to the following questions, I would recommand to suspend the publication of this paper

I would like know also why you don’t have merge the data collected by RP, US and P4 in your two experimental group.

As it was previously explained, the experiments were conducted in different years, different locations, with different animals and protocols, which we fear would increase variability.

Moreover, coud you explain why you don’t have compare the sensibility, specificity, positive and negative predictive value of RP and US diagnosis compared to P4 concentration.

According to our experimental design, we think that comparing the sensibility, specificity, positive and negative predictive value would not be the most appropriate approach. In Experiment 1, we want to check if US can improve the selection made by RP, and in Experiment 2, if the different measurements proposed can improve the selection regarding pregnancy rate. We think that, to compare SE, ES, PPV, and NPV a different experimental design would be better, by means of examination of all heifers, including the ones that did not meet the selection criteria, and perform the embryo transfer in all the animals.

Could you explain also the utility to study the possible effect of the two synchronisation protocol ?

When conducting the study using the daily work of the veterinarians, it was necessary to adapt to the protocols used. However, we believe that in order to study the potential effect of the protocols, a different experimental design from that of the current study would be necessary, as two completely independent experiments have been conducted.

Reviewer 3 Report

The MS entitled “Assessment of Luteal Function…” by Yáñez et al performed two experiments to evaluate the possibility of rectal palpation for luteal function for ET recipients, and got some meaningful results, which provide basic evidence for ET program in large animals. The MS is well prepared and performed. However, I have some questions and suggestions as below.

General comment

The criteria for selected recipients in two experiments is different? Are there any references? Pls cite them for each.

The statistics method is not complete, and the methods part is not very consistent and is not in appropriate form.

Specific comment

L26-28 the two objectives for Experiment 1 is repeated? The objective for experiment 1 is the accuracy of RP is enough

L38 In the abstract, a conclusion is needed

L40 Keywords, “P4” is needed, and “dairy cattle” is not needed 

L123 What’s IETS classification, cite references

L150 about the protocols in Experiment 1, is PGF administrated for the second one? It is not consistent with L137

L178 the formula V = 4/3πr3   is similar to L182? Because the authors used different formation

L185 When was the blood collected?

L195 what’s the CV% for ELISA?

L208 About the Embryo transfer procedure, L221-247 looks redundant, my suggestion is to provide them in the supplemental data

L256 for Pregnant rate, Chi-square test is needed as in table 6?

L280 use a subtitle to replace the original Experiment 1, the same to L311

L283 Pls provide the CL type (luteal structure present in the ovary) for these 51 animals according to the rectal palpation criteria

L286 show the original data for 44.9% as in the pregnant to the total number

L296 Table 3 add the unit for each item

L327 list the number of pregnant numbers for each item in table 6

L409 To confirm the significant difference among the Embryo preservation, which seems not consistent with L409

NONE

Author Response

Reviewer 3:

The MS entitled “Assessment of Luteal Function…” by Yáñez et al performed two experiments to evaluate the possibility of rectal palpation for luteal function for ET recipients, and got some meaningful results, which provide basic evidence for ET program in large animals. The MS is well prepared and performed. However, I have some questions and suggestions as below.

General comment

The criteria for selected recipients in two experiments is different? Are there any references? Pls cite them for each.

The references used were:

Zemjanis, R. Diagnostic and Therapeutic Tecniques in Animal Reproduction; The Williams and Wilkins Company: Baltimore, 1970 (Experiment 1)

Yáñez, U.; Murillo, A. V.; Becerra, J.J.; Herradón, P.G.; Peña, A.I.; Quintela, L.A. Comparison between Transrectal Palpation, B-Mode and Doppler Ultrasonography to Assess Luteal Function in Holstein Cattle. Front Vet Sci 2023, 10, doi:10.3389/fvets.2023.1162589 (Experiment 2). This is a published article by our group, and the references for the criteria are:

Donate J, Quintela L, Díaz C, Becerra J, Herradón P. Eficacia de la palpación rectal y la ecografía en la determinación de las fases del ciclo estral en la vaca de leche. I Congreso Internacional AERA-BAS. Gijón (2008)

Bicalho RC, Galvão KN, Guard CL, Santos JEP. Optimizing the accuracy of detecting a functional corpus luteum in dairy cows. Theriogenology (2008) 70:199–207. doi: 10.1016/j.theriogenology.2008.03.015

The statistics method is not complete, and the methods part is not very consistent and is not in appropriate form.

The statistics section was modified. Additional information about the logistic regression was included in Experiment 2 to make it clearer.

Specific comment

L26-28 the two objectives for Experiment 1 is repeated? The objective for experiment 1 is the accuracy of RP is enough

It is not exactly the same. One of the objectives was to check the accuracy of RP to select recipients by the evaluation of the CL, and compared it to US and P4, that is, just the ability to select recipients with a functional CL. The second objective was to verify if US or P4 could offer additional information to improve pregnancy rates.

L38 In the abstract, a conclusion is needed

The abstract was modified according to the reviewer’s suggestions.

L40 Keywords, “P4” is needed, and “dairy cattle” is not needed

The keywords were modified according to the reviewer’s suggestions.

L123 What’s IETS classification, cite references

A reference was included as suggested.

L150 about the protocols in Experiment 1, is PGF administrated for the second one? It is not consistent with L137

The Figure 1 was corrected.

L178 the formula V = 4/3πr3   is similar to L182? Because the authors used different formation

The first formula (V=4/3πr^3) was the one used to calculate the volume of the CL, assuming a perfect sphere. However, the second formula (V=4/3 πabc) was the one used to calculate the volume of the ovary, assuming an ellipse.

L185 When was the blood collected?

Samples were collected at the end of the examination to select the recipients. The manuscript was modified so it would be clearer.

L195 what’s the CV% for ELISA?

According to the manufacturer, the inter – assay variation was determined by repeated measurements of samples with 3 different kit lots, obtaining the following results:

Sample

N

Mean (ng/mL)

CV(%)

1

18

1.2

7.2

2

18

38.7

3.1

L208 About the Embryo transfer procedure, L221-247 looks redundant, my suggestion is to provide them in the supplemental data

We have no preference about including this information in the manuscript or as supplemental data. If the other reviewers and the editor agree, we can modify the manuscript in the next revision and include the embryo transfer procedure as supplemental materials.

L256 for Pregnant rate, Chi-square test is needed as in table 6?

In Table 6, the results for the logistic regression are displayed, including descriptive statistics and significance. If necessary, we can add the OR of all variables (in addition to the ones already mentioned in the text) to complete the information, if that is what the reviewer means.

L280 use a subtitle to replace the original Experiment 1, the same to L311

We are not sure about what the reviewer means with this suggestion.

L283 Pls provide the CL type (luteal structure present in the ovary) for these 51 animals according to the rectal palpation criteria

The manuscript was modified according to the reviewer’s suggestions.

L286 show the original data for 44.9% as in the pregnant to the total number

We are not sure about what the reviewer means with this suggestion.

L296 Table 3 add the unit for each item

The table was modified according to the reviewer’s suggestions.

L327 list the number of pregnant numbers for each item in table 6

The table was modified according to the reviewer’s suggestions.

L409 To confirm the significant difference among the Embryo preservation, which seems not consistent with L409

We are not sure about what the reviewer means with this suggestion. We did not observe a statistically significant difference regarding the variable embryo preservation.

Round 2

Reviewer 2 Report

Thanks for your comment.

I have no more remarks.

Author Response

We appreciate the reviewer's comments, which have undoubtedly helped to improve the manuscript.